# Timing of ESCRT-III protein recruitment and membrane scission during HIV-1 assembly

**Daniel S Johnson[1,2], Marina Bleck[1], Sanford M Simon[1]\***

[1]Laboratory of Cellular Biophysics, The Rockefeller University, New York, United States; [2]Department of Physics and Astronomy, Hofstra University, Hempstead, United States

**Abstract** The Endosomal Sorting Complexes Required for Transport III (ESCRT-III) proteins are critical for cellular membrane scission processes with topologies inverted relative to clathrin-mediated endocytosis. Some viruses appropriate ESCRT-IIIs for their release. By imaging single assembling viral-like particles of HIV-1, we observed that ESCRT-IIIs and the ATPase VPS4 arrive after most of the virion membrane is bent, linger for tens of seconds, and depart ~20 s before scission. These observations suggest that ESCRT-IIIs are recruited by a combination of membrane curvature and the late domains of the HIV-1 Gag protein. ESCRT-IIIs may pull the neck into a narrower form but must leave to allow scission. If scission does not occur within minutes of ESCRT departure, ESCRT-IIIs and VPS4 are recruited again. This mechanistic insight is likely relevant for other ESCRT-dependent scission processes including cell division, endosome tubulation, multivesicular body and nuclear envelope formation, and secretion of exosomes and ectosomes.
DOI: https://doi.org/10.7554/eLife.36221.001

## Introduction

ESCRTs are categorized into groups −0 through -III and act in various cellular processes including cell division, multivesicular body formation, and wound repair (*Hurley, 2015*). An investigation of protein sorting between endosomes and lysosomes initially led to the discovery of ESCRT-I (*Katzmann et al., 2001*), with other ESCRTs identified soon afterwards (*Babst et al., 2002a*, *2002b*; *Katzmann et al., 2003*). Sequential recruitment of the ESCRTs enables division of membrane compartments, with ESCRT-III being critical for the membrane scission process (*Henne et al., 2013*). ESCRT-IIIs have a five alpha-helix core structure and assemble in vitro into macromolecular rings or spirals (*McCullough et al., 2015*; *Muzioł et al., 2006*). ESCRT-IIIs are believed to polymerize in the neck of the membrane constriction to drive membrane fission (*Cashikar et al., 2014*; *Dobro et al., 2013*; *Fabrikant et al., 2009*; *Hanson et al., 2008*; *Henne et al., 2012*; *Lata et al., 2008*; *McCullough et al., 2015*).

Scission of the membrane, even in the presence of ESCRT-IIIs, stalls in the absence of the hexameric AAA$^+$ ATPase (*Morita et al., 2010*), VPS4A/B, which contains a microtubule interacting and transport (MIT) domain that binds to MIT interacting motifs (MIM) of ESCRT-IIIs (*Obita et al., 2007*; *Stuchell-Brereton et al., 2007*). The final scission process is believed to be associated with VPS4 working on ESCRT-IIIs, but the mechanism is still unresolved. In some models, the ESCRT-IIIs provide the motive force for scission and the VPS4 is required after scission to recycle the ESCRT-IIIs for subsequent scissions (*Lata et al., 2008*; *Wollert et al., 2009*). In other models, the VPS4 is actively engaging the ESCRT-IIIs prior or during scission by actively remodeling ESCRT-IIIs in order to force scission (*Saksena et al., 2009*), by rearranging ESCRT-IIIs as part of the pathway toward scission

\*For correspondence:
Sanford.Simon@rockefeller.edu

**Competing interests:** The authors declare that no competing interests exist.

**eLife digest** Viruses need to be able to enter and leave the cells of their hosts to multiply and spread infections. Once inside the cell, many viruses, including the HIV virus, hijack the cell's genetic material to produce more HIV particles and release them back into the surroundings. As the new viruses leave the cell, they wrap themselves in the membrane of their host cell.

The shell of HIV consists of thousands of copies of a protein called Gag, which helps to release the viruses. Gag aggregates inside of the host cell membrane, which then begins to bulge outwards forming a spherical bud that subsequently pinches off. These released virus particles are now able to infect other cells.

Both assembling and budding of the virus particles requires the help of specific cell proteins called ESCRTs. Cells usually use ESCRTs to cut membrane-bound compartments during cell division. HIV can control ESCRTs to separate the bud from the cell, but where and how they are involved is still not fully understood. Some models propose that the ESCRTs bend the membrane, while others suggest that the ESCRTs are required during the split, providing the force for the cutting process.

Now, Johnson et al. used fluorescent light to follow how individual viruses in human cells grown in the laboratory assemble, and polarized light to detect how the orientation of Gag changed as the membrane bends into a sphere. The experiments revealed that the virus particles started to form a sphere at the same time Gag began to gather at the cell membrane, before any ESCRTs were present. Once enough Gag proteins had accumulated, the ESCRTs were recruited to the spot and removed within tens of seconds, just before the virus was cut off from the host cell. This suggest that the ESCRTs may help prepare the membrane to split, but are not involved in the actual cutting process.

The findings of Johnson et al. will help us to better understand the behavior of HIV viruses. Knowing how they assemble and leave the host cell may help to create new strategies for disrupting these processes in particular, so stopping the virus from spreading. Moreover, ESCRTs are involved in numerous activities of a cell, such as cell division or repair of the cell surface after rupture, but little is known about how they work. For the first time, it has been shown that ESCRTs are neither the driving force for bending the membrane nor are they present when the cutting of a virus happens. This may also apply to other processes in the cell that involve membrane-splitting by ESCRT proteins, and researchers may be compelled to reevaluate their proposed role in other systems.

DOI: https://doi.org/10.7554/eLife.36221.002

(*Cashikar et al., 2014*), or by binding to ESCRT dome structures in order to add rigidity necessary for scission (*Fabrikant et al., 2009*).

ESCRT complexes are hijacked by HIV to enable separation of the viral particle from the host cell plasma membrane. The production of enveloped HIV-1 at the plasma membrane occurs with the recruitment of the structural protein Gag at individual assembly sites. The carboxyl terminus of Gag has a motif that is essential for recruitment of ESCRTs. First, Gag recruits the 'early' factors like ESCRT-I/TSG101 and ALIX which contribute to subsequent recruitment of ESCRT-III proteins. The ESCRT-IIIs then polymerize into structures that are believed to constrict the neck and drive membrane fission. HIV release appears to require fewer members of the ESCRT family than other processes. Redundancy likely makes many variants, such as CHMP5, CHMP6 and CHMP7, only conditionally necessary (*Morita et al., 2011*). ESCRT-IIIs that are essential for assembly of HIV-1 include CHMP2 (either A or B variant) and CHMP4B, which are recruited to site of budding with other proteins such as ESCRT-I/TSG101 and ALIX which interact with Gag (*Morita et al., 2011*). The reduced number of required ESCRTs makes HIV assembly an approachable system for studying the biophysics of ESCRT-mediated membrane scission.

Prior to viral particle separation from a host cell, a roughly spherical particle is formed (*Martin-Serrano et al., 2003*), but the topological pathway and timing of events to reaching the Gag sphere has not previously been followed in vivo. The order of some of the events in virion assembly has been resolved by live-cell microscopy. First, the HIV- genome is recruited to the membrane, potentially with a few Gag molecules (*Jouvenet et al., 2009*). Then, over a 5–30 min period, the Gag

accumulates around the genome (*Ivanchenko et al., 2009*; *Jouvenet et al., 2008*). Once Gag reaches a steady-state, ESCRT-III and VPS4 are transiently recruited at the site of assembly (*Baumgärtel et al., 2011*; *Bleck et al., 2014*; *Jouvenet et al., 2011*). The timing of some critical steps is not known impacting our understanding of the mechanism. It is not known whether bending occurs before, during, or after the transient recruitment of ESCRTs. Thus, is membrane bending driven by Gag multimerization, by Gag engagement with the HIV-1 genome or by the ESCRTs? It is also not known if scission occurs before, during, or after the transient recruitment of ESCRT-IIIs or VPS4. Do they generate the force for scission, do they prepare the membrane for scission, or does VPS4 recycle ESCRT-IIIs after scission?

   Here, we investigated, during the assembly of HIV Gag, the temporal recruitment of ESCRT-III proteins and VPS4 relative to membrane scission. We also examined membrane curvature during Gag assembly to determine when a spherical particle forms relative to ESCRT-III recruitment. We find that membrane bending occurs contemporaneous with recruitment of Gag and prior to arrival ESCRT-III. The ESCRT-IIIs and the VPS4 ATPase arrive after Gag assembly has concluded, remain at the membrane for tens of seconds, and then leave tens of seconds before scission. During the period after departure of the ESCRT-IIIs, neutralizing the surface charge on the membrane accelerates the membrane scission.

## Results

### ESCRT-IIIs appear and disappear from site of virus-like particle assembly prior to scission

To determine the timing of ESCRT recruitment relative to membrane bending and scission, we quantified ESCRT recruitment during the assembly of HIV-1 virus like particles (VLPs) while assaying membrane bending and scission. Scission was assayed by monitoring the ability of protons to flow between the cytosol and the lumen of the virion. The pH in the lumen of the virion was monitored with a pH-sensitive GFP (pHlourin) (*Miesenböck et al., 1998*) fused to Gag (*Jouvenet et al., 2008*) while modulating the cytoplasmic pH by cycling the $pCO_2$ every 10 s between 0 and 10%, thus an average of 5% $pCO_2$ (*Figure 1—figure supplement 1*). $CO_2$ rapidly diffuses across plasma membranes (*Hulikova and Swietach, 2014*; *Simon et al., 1994*) and is converted to carbonic acid by cytoplasmic carbonic anhydrase, altering the cytoplasmic pH. We have previously observed that Gag-pHlourin in a budded VLP is less sensitive to changes of $pCO_2$ than in the cytosol, suggesting carbonic anhydrase is excluded from VLPs (*Jouvenet et al., 2008*).

   At sites of VLP assembly, the average Gag-pHluorin fluorescence increase was similar to the increase of Gag-mEGFP (*Jouvenet et al., 2008*), but the intensity oscillated in sync with switching the $pCO_2$ (*Figure 1A*, *Figure 1—figure supplement 2*, *Figure 1—video 1* and *Figure 1—video 2*). At various times after Gag accumulation reached a steady-state maximum the magnitude of oscillations decreased, indicating a loss of the ability of protons to move between the VLP and cytoplasm due to scission. $CO_2$ may still cross the VLP membrane after scission and enter via gaps in the lattice of the immature Gag lattice (*Briggs et al., 2009*; *Carlson et al., 2008*; *Woodward et al., 2015*; *Wright et al., 2007*). However, the greatly reduced sensitivity to pH indicates carbonic anhydrase is not present, consistent with analysis of cellular proteins in HIV particles by mass spectrometry (*Ott, 2008*). Not surprisingly, scission was never observed before Gag had finished accumulating at individual VLPs. Unexpectedly, the ESCRT-III CHMP4B both appeared (Avg = 59 s, N = 30 out of 30) and disappeared (Avg = 22 s, N = 27 out of 30) prior to scission (*Figure 1B*).

   CHMP4B has been proposed to form a circular/spiral structure which supports assembly of a smaller CHMP2(A/B) dome which generates fission by pulling the neck together (*Fabrikant et al., 2009*). Thus, it is possible that CHMP4B may be removed prior to scission leaving CHMP2A or CHMP2B present to facilitate fission. To probe the timing of CHMP2, endogenous CHMP2A and CHMP2B were lowered with siRNA (*Figure 1—figure supplement 3*) to facilitate observation of mCherry-CHMP2A or mCherry-CHMP2B (*Figure 1C and E*, *Figure 1—figure supplement 2*). CHMP2A and CHMP2B both appeared (Avg. = 77 s, N = 29 out of 29; and 63 s, N = 23 out of 23, respectively) and disappeared prior to scission (Avg. = 23 s, N = 26 out of 29; and 27 s, N = 23 out of 23, respectively *Figure 1D,F*). This observation indicates their assembly and disassembly is also not physically forcing scission.

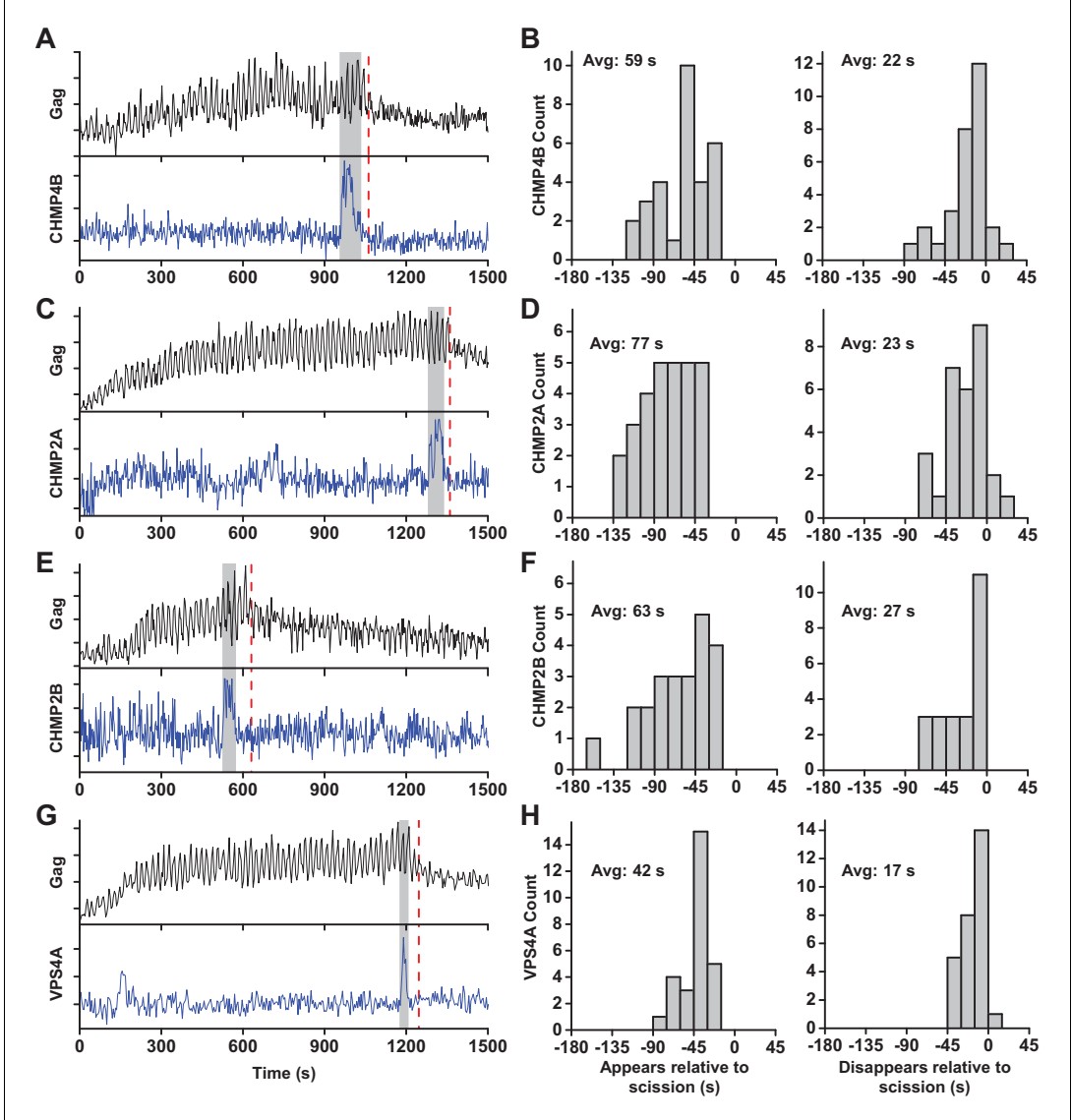

**Figure 1.** ESCRT-IIIs and VPS4A transiently recruited prior to scission. (A) Example trace of Gag-pHluorin assembling into single VLP while the $pCO_2$ in the imaging media was repeatedly switched between 0% and 10% every 10 s. Moment of scission is indicated by red dashed line. CHMP4B-mCherry was temporarily recruited (indicated by grey zone) to the site of VLP assembly following the loss of pH modulation sensitivity. (B) Histograms of appearance and disappearance of CHMP4B prior to scission. (C-H) Example traces and histograms of appearance and disappearance, relative to scission of the VLP, for mCherry-CHMP2A (C and D), mCherry-CHMP2B (E and F) and mCherry-VPS4A (G and H).
DOI: https://doi.org/10.7554/eLife.36221.003

The following video and figure supplements are available for figure 1:

**Figure supplement 1.** Flow chamber configuration for ESCRT-III assisted membrane scission studies.
DOI: https://doi.org/10.7554/eLife.36221.004

**Figure supplement 2.** Example traces of scission relative to recruitment of ESCRT-III or VPS4A.
DOI: https://doi.org/10.7554/eLife.36221.005

**Figure supplement 3.** Knockdown of CHMP2A or CHMP2B by siRNA.
DOI: https://doi.org/10.7554/eLife.36221.006

**Figure supplement 4.** CHMP4B was recruited prior to VPS4A.
DOI: https://doi.org/10.7554/eLife.36221.007

**Figure 1—video 1.** Gag-pHluorin (left side of video) and mCherry-CHMP4B (right side) were imaged while the $CO_2$ in the media was modulated between 0 and 10% every 10 s.
DOI: https://doi.org/10.7554/eLife.36221.008

*Figure 1 continued on next page*

*Figure 1 continued*

**Figure 1—video 2.** Example of individual puncta of Gag-pHluorin assembly (left side, $CO_2$ switching every 10 s) with mCherry-CHMP4B (right side) recruitment.

DOI: https://doi.org/10.7554/eLife.36221.009

**Figure 1—video 3.** Gag-pHluorin (top of video) and mCherry-VPS4A (bottom) were imaged while the $CO_2$ in the media was modulated between 0 and 10% every 10 s.

DOI: https://doi.org/10.7554/eLife.36221.010

**Figure 1—video 4.** Example of individual puncta of Gag-pHluorin assembly (left side, $CO_2$ switching every 10 s) with mCherry-VPS4A (right side) recruitment.

DOI: https://doi.org/10.7554/eLife.36221.011

Next the dynamics of recruitment of VPS4, the energy providing ATPase, was monitored (*Figure 1G* and *Figure 1—figure supplement 2*, *Figure 1—video 3* and *Figure 1—video 4*). Similar to the ESCRT-IIIs, VPS4A also appeared prior to scission (Avg = 42 s, N = 28 out of 28) and

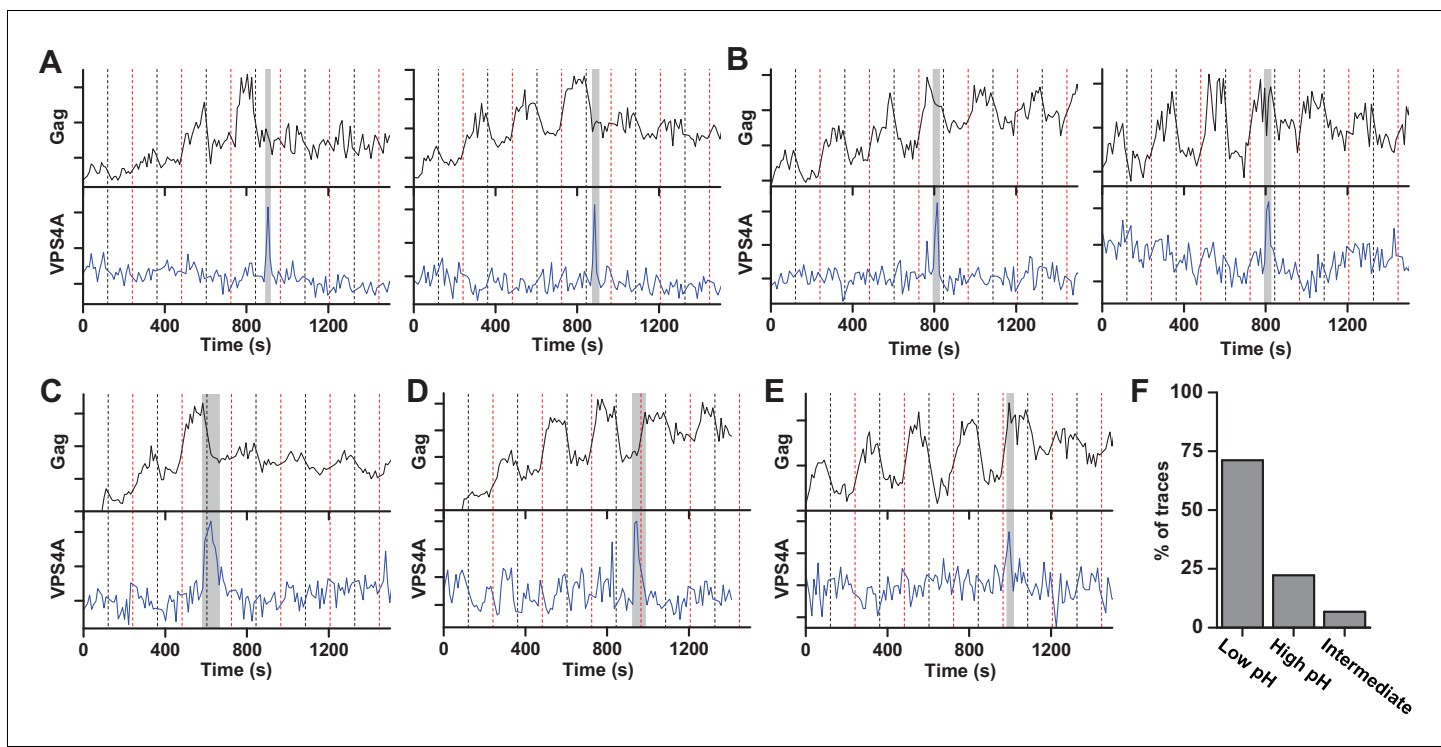

**Figure 2.** Scission more likely at lower cytoplasmic pH. (**A**) Example traces of Gag-pHluorin assembly while $pCO_2$ was switched every 120 s between 0% (red dashed line, greater fluorescence emission) and 10% (black dashed line, lower fluorescence emission). In these traces the fluorescence intensity became fixed in the low pH state (10% $pCO_2$) after reaching an assembly plateau. VPS4A appeared and disappeared during the first low pH state. (**B**) Examples in which fluorescence intensity became fixed in the high pH state (0% $pCO_2$). VPS4A appeared and disappeared during the first trapped high pH state. (**C**) Example in which fluorescence became fixed in low pH state. VPS4A disappeared during the first trapped low pH state, but appeared during the previous high pH state. (**D**) Example in which fluorescence became fixed in high pH state. VPS4A disappeared during the first trapped high pH state, but appeared during the previous low pH state. (**E**) Example trace in which fluorescence intensity because fixed in an intermediate state. (**F**) Bar graph of cytoplasmic pH condition in which scission occurs (N = 45). Scission is ~3-fold more likely at low pH (10% $pCO_2$) compared to high pH (0% $pCO_2$) condition. A small percentage of VLPs were trapped in an intermediate state.

DOI: https://doi.org/10.7554/eLife.36221.012

The following videos are available for figure 2:

**Figure 2—video 1.** Gag-pHluorin (left side of video) and mCherry-VPS4A (right side) were imaged while the $CO_2$ in the media was modulated between 0 and 10% every 120 s.

DOI: https://doi.org/10.7554/eLife.36221.013

**Figure 2—video 2.** Example of individual puncta of Gag-pHluorin assembly (left side, $CO_2$ switching every 120 s) with mCherry-VPS4A (right side) recruitment.

DOI: https://doi.org/10.7554/eLife.36221.014

disappeared prior to scission (Avg = 17 s, N = 27 out of 28, *Figure 1H*). On average VPS4A disappeared from the assembly site closer to the time of scission than CHMP4B (5 s), CHMP2A (6 s) or CHMP2B (10 s). A simultaneous measurement of CHMP4B and VPS4A confirmed CHMP4B was recruited ~5 s prior to VPS4A (N = 41) (*Figure 1—figure supplement 4*), which agrees with previous results in HeLa cells (*Bleck et al., 2014*).

## Acidification of cytosol accelerates scission

Our results indicate that the ESCRT-IIIs and the ATPase VPS4 leave the membrane prior to scission. It is possible that the ESCRT-IIIs play an essential role in tightening the membrane neck, but then need to be cleared away to allow for the opposing membranes to come closer for the scission reaction. The specific lipid composition in the neck is not known. However, both HIV-1 Gag and the ESCRT complexes are recruited to regions rich in phosphatidylinositol-4,5-bisphosphate ($PIP_2$), which has four negative charges at pH 7 (*Kooijman et al., 2009*), with the net negative charge being the critical parameter for engaging ESCRTs (*Lee et al., 2015*). Thus, ESCRTs at the neck may bias the composition toward more negatively charged lipids (*Chiaruttini et al., 2015*). The $pK_a$ values for the two phosphate groups on the inositol of $PIP_2$ are 6.5 and 7.7 (*van Paridon et al., 1986*). The change in pHluorin intensity from 0 and 10% $pCO_2$, indicates the cytosolic pH ranges from ~7.5 to ~6.5. Therefore, lowering the pH by raising the $pCO_2$ to 10% should protonate and reduce the charge of these negatively charged lipids thus reducing the repulsive force between the membranes. To test if

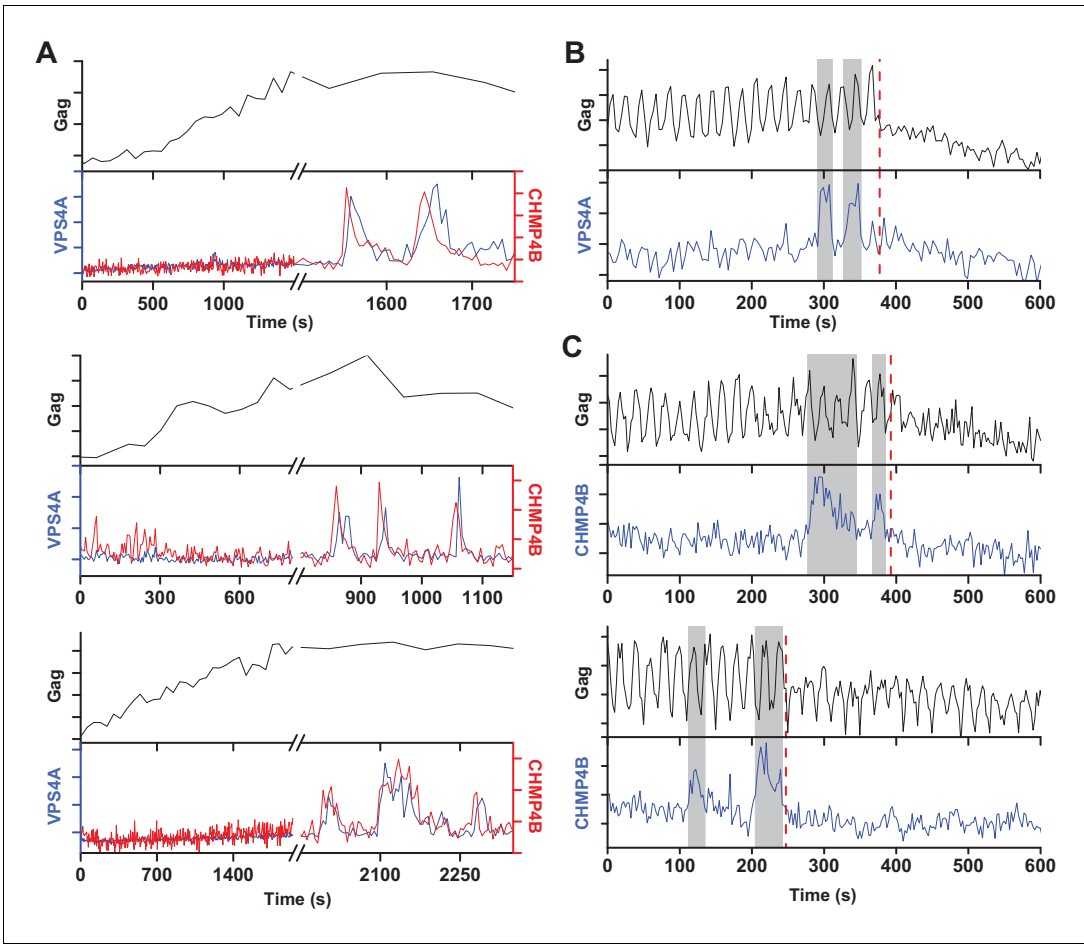

**Figure 3.** A new round of ESCRT-III recruitment required following failed scission event. (**A**) Example traces of multiple waves of VPS4A and CHMP4B recruited following cessation of Gag accumulation. (**B**) Example trace of multiple recruitments of VPS4A prior to scission (red dashed line). (**C**) Example traces of multiple recruitments of CHMP4B prior to scission (red dashed line).
DOI: https://doi.org/10.7554/eLife.36221.015

scission was affected by changes in cytosolic pH, we switched the $pCO_2$ at a slower rate, every 120 s (*Figure 2*, *Figure 2—video 1* and *Figure 2—video 2*). Scission was ~3X more likely when the cytoplasm was in the low pH state (10% $pCO_2$) than the high pH state (0% $pCO_2$) (*Figure 2F*) consistent with the idea that scission is more likely when the net negative charge on phospholipids in the viral neck is reduced by protonation. Other common phospholipids like phosphatidic acid (PA), phosphatidylethanolamine (PE), phosphatidylcholine (PC), phosphatidylserine (PS), and phosphatidylglycerol (PG), have $pK_a$ values outside the pH range used in these experiments so are not expected to be as sensitive to charge modulation during pH switching as $PIP_2$. However, other lipids or components with pH-dependent charge sensitivity might also or alternatively result in these observations.

## Additional rounds of ESCRT-III/VPS4 recruitment occur following failed scission

Multiple rounds of ESCRT-III/VPS4 recruitment were previously observed following completion of Gag accumulation (*Baumgärtel et al., 2011*; *Jouvenet et al., 2011*). It is possible that the first wave of ESCRT-III/VPS4 led to productive scission and the subsequent rounds are inconsequential. Alternatively, the initial waves could be non-productive, perhaps a consequence of a failure to recruit both ESCRT-IIIs and VPS4A concurrently, necessitating additional rounds. While simultaneously imaging Gag-pHluorin, CHMP4B and VPS4A, when there were multiple waves, both CHMP4B and VPS4A were recruited (*Figure 3*). Multiple waves of recruitment were observed in ~20% of traces, consistent with our previous observations of repeat rounds of recruitment of CHMP1B, CHMP4B, CHMP4C and VPS4A during assembly of HIV-1 and EIAV (*Jouvenet et al., 2011*). Scission was only observed after the final wave of recruitment of ESCRT-III/VPS4 (*Figure 3B,C*). Thus, not every cycle of arrival and then dispersal of ESCRT-III/VPS4 leads to subsequent scission. If scission does not occur then a subsequent cycle of recruitment and dispersal of the ESCRT-III/VPS4 is required to complete the process. If the function of VPS4A is to recycle ESCRT-IIIs after scission then only a single wave would be expected since ESCRT-IIIs would not be removed until after the single scission event has occurred.

## Membrane bending occurs throughout assembly of virus-like particle

Next, we set out to determine when membrane bending occurs relative to the assembly of Gag and recruitment of ESCRT-IIIs. We expressed a fluorescent protein (either EGFP or one of two circularly permutated superfolder variants, sf3 or sf11 (*Pédelacq et al., 2006*) as a fusion to Gag (at the carboxyl terminus, p6-GFP, or in the matrix protein of Gag, MA-sf3 or MA-sf11) to be able to follow membrane bending in live-cell imaging via changes in anisotropy of the GFP tag.

The orientation of the chromophore was characterized with a custom built polarized total internal reflection fluorescence (TIRF) illuminator (*Johnson et al., 2014*). During accumulation of Gag at VLPs, the emission of the GFP was quantified while excitation alternated with polarization perpendicular ($\hat{p}$ polarized) followed by parallel ($\hat{s}$ polarized) to the glass surface. Orientation was characterized by the ratio of emission intensities (P/S) and total Gag was monitored by P+2S (*Figure 4A* and *Figure 4—figure supplement 1*, *Figure 4—video 1* and *Figure 4—video 2*) (*Anantharam et al., 2010*). As Gag accumulated, the ratio of P/S dropped from ~2 to ~1.4, with little variation (±0.1) between the Gag-GFP versions (*Figure 4B*). The drop in P/S correlated with the increase in Gag as would be expected if the plasma membrane was bending during Gag assembly. The halfway decrease of P/S occurred prior to the halfway increase of the total Gag fluorescence (*Figure 4C*). Following Gag recruitment, as indicated by a plateau in Gag signal, there was no transition in P/S. This observation is inconsistent with the subsequent recruitment of the ESCRT-IIIs facilitating the transition from a flat lattice to a spherical particle.

Bending was also investigated during the assembly of a Gag that is missing its carboxyl terminal p6 domain which functions to recruit early acting proteins like ESCRT-I/TSG101 or ALIX. A similar drop in P/S during assembly of Gag was observed with Gag-Δp6 (*Figure 4A*, *Figure 4—figure supplement 2*) indicating that the ESCRTs recruited via p6 are also not necessary for the transition from a flat lattice to a spherical particle.

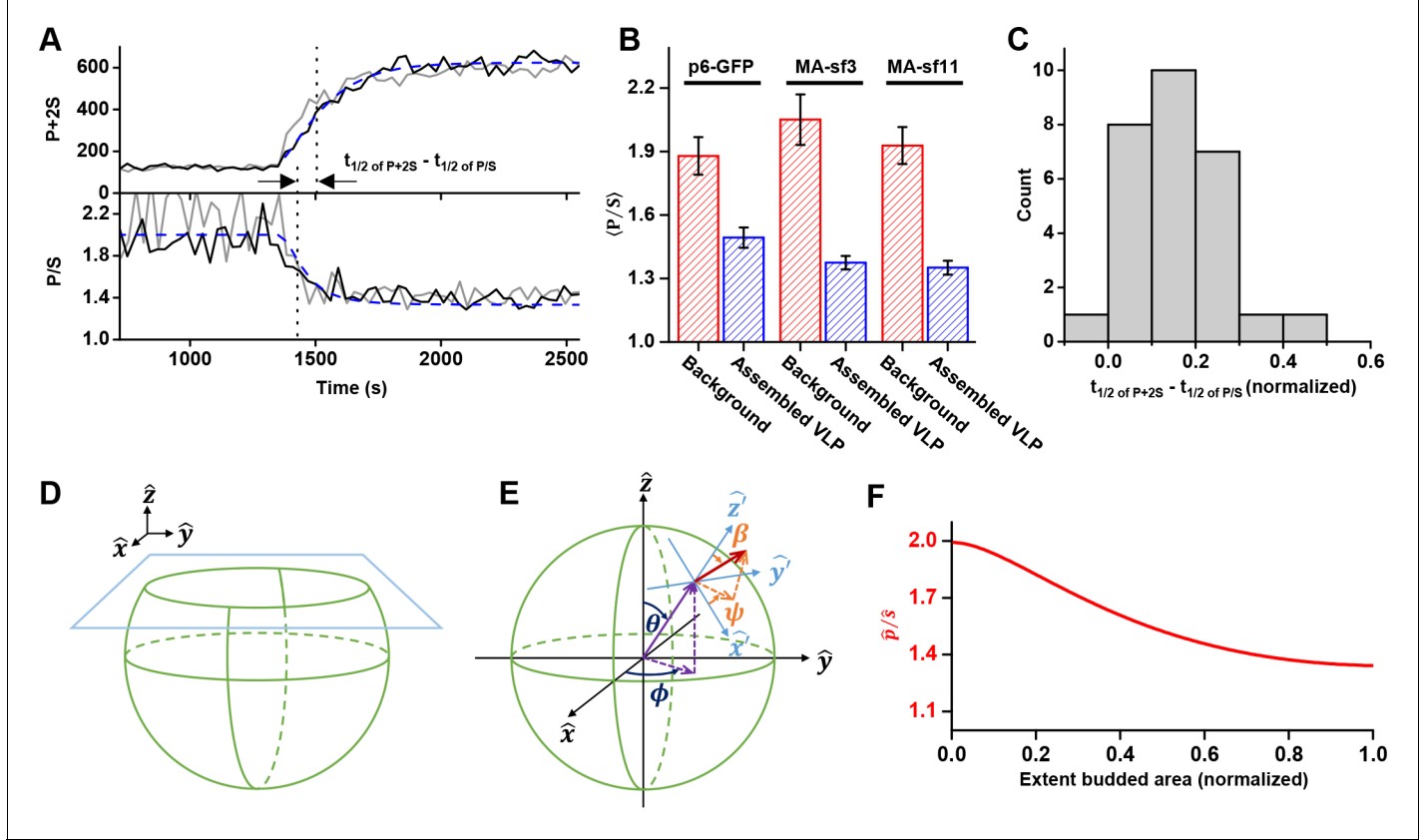

**Figure 4.** Structural changes in VLPs throughout Gag accumulation. (**A**) Example traces of wild-type Gag-GFP (black line, sf3 in Matrix of Gag) and Gag-Δp6 (grey line, missing ESCRT-I recruiting p6 domain) assembling into single VLPs. Images were collected every 30 s with excitation illumination polarized either perpendicular ($\hat{p}$) or parallel ($\hat{s}$) to the glass surface. Total Gag characterized by P+2S (top) and relative average dipole orientation by P/S (bottom). P+2S from wild-type Gag was fit to an exponential and used to predict an expected P/S (blue dashed line) assuming membrane bending throughout assembly. (**B**) Comparison of average P/S from all traces before VLP assembly (membrane background) and after VLP assembly (plateau region) for three different tagged versions of wild-type Gag. p6-GFP (N = 8), MA-sf3 (N = 9), and MA-sf11 (N = 7). Error bars represent s.d. (**C**) To compare the evolution of VLP structure to the assembly of Gag the time for each assembly trace was normalized from 0 (beginning of Gag assembly) to 1 (end of Gag assembly). A normalized time difference for each trace between Gag half assembly [½ (P+2S)$_{max}$] and the dipole half drop [½ (P/S)$_{max}$] was found and all normalized differences were compiled into a histogram. (**D**) Illustration of sphere budding from flat membrane. (**E**) Illustration of coordinate system with $\theta$ and $\phi$ representing position on the sphere and $\beta$ and $\psi$ representing orientation of excitation dipole. (**F**) Predicted P/S when $\beta$ = 45°, background intensity is 45% of full assembly intensity, and evanescent field penetration depth is the same as the radius of the VLP.
DOI: https://doi.org/10.7554/eLife.36221.016

The following video and figure supplements are available for figure 4:

**Figure supplement 1.** Example traces of Gag accumulation (quantified as P+2S) and fluorophore polarization (quantified as P/S) during VLP assembly.
DOI: https://doi.org/10.7554/eLife.36221.017

**Figure supplement 2.** Example traces of Gag-Δp6-mEGFP accumulation (quantified as P+2S) and fluorophore polarization (quantified as P/S) during VLP assembly.
DOI: https://doi.org/10.7554/eLife.36221.018

**Figure 4—video 1.** Images of Gag-mEGFP were acquired with excitation perpendicular (P) or parallel (S) to the glass surface.
DOI: https://doi.org/10.7554/eLife.36221.019

**Figure 4—video 2.** Example of individual puncta of Gag-mEGFP with P/S and P+2S images.
DOI: https://doi.org/10.7554/eLife.36221.020

## A simulation of spherical budding reproduced the time course of the P/S ratio relative to P+2S

In order to better understand the observed P/S ratio, we formulated an expected P/S ratio for a spherical cap growing out of a flat membrane (**Figure 4D**). Briefly, we assumed the growing bud consisted of excitation dipoles uniformly distributed across the surface, with the dipoles oriented an

angle $\beta$ relative to the surface normal. A predicted P/S with respect to $\beta$ and the normalized budded surface area (area from 0 to 1) was then found by integrating over all defined dipole orientations and the extent budded surface area.

More specifically, using coordinates described previously (*Anantharam et al., 2010*), position on the surface of the sphere was given in terms of a polar angle $\theta$ and an azimuthal angle $\phi$, and the current extent budded was defined by $\theta$ (*Figure 4E*). Thus, when $\theta = 0°$ there was no budding, when $\theta = 90°$ the sphere was half budded with dipoles from $\theta = 0° \rightarrow 90°$, and when $\theta = 180°$ the sphere was fully budded with dipoles from $\theta = 0° \rightarrow 180°$. A uniform distribution of excitation dipoles was assumed on the bud (no dependence on $\phi$ or $\theta$); however, at any given position these dipoles had an angular distribution that depended on the polar angle $\beta$ relative to the surface normal: $\rho(\beta)$. For instance, if all dipoles were oriented at $\beta = 45°$ then $\rho(\beta) = \delta(\beta - 45°)$ where $\delta(x)$ is a delta function. The angular distribution was assumed to be uniform relative to the surface azimuthal angle $\psi$ and the sphere was assumed to be smaller than the optical resolution of the microscope.

A predicted P/S ($\rho(\beta)$) relative to extent budded $\theta$ was found by determining the average component of the dipole excitation in $\hat{y}$ (parallel to glass surface) and in $\hat{z}$ (normal to glass surface). Note: Experimentally due to azimuthal scanning we excited in both $\hat{x}$ and $\hat{y}$, each 50% the time, but for simplicity in this analysis 100% excitation in $\hat{y}$ was assumed since $\hat{x}$ and $\hat{y}$ are symmetric. The total collected fluorescence, S and P, in $\hat{y}$ and $\hat{z}$ were predicated by:

$$\mathrm{S} = \int_0^\theta \int_0^{2\pi} \int_0^\pi \int_0^{2\pi} \mathrm{Q}_{||} \mid \mathrm{E}_{\hat{y}} \mu_{\hat{y}} \mid^2 \sin(\theta)\sin(\beta)\,d\psi\,d\beta\,d\phi\,d\theta \tag{1}$$

$$\mathrm{P} = \int_0^\theta \int_0^{2\pi} \int_0^\pi \int_0^{2\pi} \mathrm{Q}_\perp \mid \mathrm{E}_{\hat{z}} \mu_{\hat{z}} \mid^2 \sin(\theta)\sin(\beta)\,d\psi\,d\beta\,d\phi\,d\theta \tag{2}$$

where $\mu_{\hat{y}}$ and $\mu_{\hat{z}}$ are the components of the excitation dipole in $\hat{y}$ and $\hat{z}$ with respect to positon on the bud surface, $\mathrm{E}_{\hat{y}}$ and $\mathrm{E}_{\hat{z}}$ are the excitation electric field components in $\hat{y}$ and $\hat{z}$, and $\mathrm{Q}_{||}$ and $\mathrm{Q}_\perp$ are the light collection efficiencies of the microscope objective for dipoles parallel and perpendicular to the glass surface. The excitation field intensity in $\hat{y}$ and $\hat{z}$ were assumed to be the same, though this was an approximation since in reality $\hat{p}$ had a small component of $\hat{s}$ (*Sund et al., 1999*). In addition, in TIR the excitation field decayed exponentially with distance from the glass surface, $\mathrm{E}_{\hat{y}\,\mathrm{or}\,\hat{y}} = \mathrm{E}e^{-z/2d} = \mathrm{E}e^{-(1-\cos(\theta))/2\mathrm{d}}$ where the characteristic decay constant was defined in terms of a fraction of the radius of the VLP. The collection efficiency for emission parallel $\mathrm{Q}_{||}$ versus perpendicular $\mathrm{Q}_\perp$ were also assumed to be the same ($\mathrm{Q}_{||} = \mathrm{Q}_\perp$), which was an approximation based on the use of a high numerical aperture objective (*Anantharam et al., 2010*). From coordinate transforms described previously (*Anantharam et al., 2010*; *Sund et al., 1999*), the components of the dipoles in $\hat{y}$ and $\hat{z}$ are given by:

$$\mu_{\hat{y}} = \rho(\beta)[\cos(\theta)\sin(\phi)\sin(\beta)\cos(\psi) + \cos(\phi)\sin(\psi)\sin(\beta) + \sin(\theta)\sin(\phi)\cos(\beta)] \tag{3}$$

$$\mu_{\hat{z}} = \rho(\beta)[-\sin(\theta)\sin(\beta)\cos(\psi) + \cos(\theta)\cos(\beta)] \tag{4}$$

$\mathrm{P}/S_{\mathrm{VLP}}(\theta, \rho(\beta))$ was solved computationally (Mathematica, Wolfram) with $\theta$ parameterized in terms of normalized surface area ($\mathrm{A} : 0 \rightarrow 1$) as $\theta = \cos^{-1}(1 - 2\mathrm{A})$. In addition, contribution from fluorescence outside of the puncta was accounted for as follows:

$$\mathrm{P}/S(\mathrm{A}, \rho(\mathrm{A}), \mathrm{C}_{\mathrm{back}}) = \frac{\mathrm{A} \cdot \mathrm{P}/S_{\mathrm{VLP}}(\mathrm{A}, \rho(\mathrm{A})) + \mathrm{C}_{\mathrm{back}} \cdot \mathrm{P}/S_{\mathrm{Background}}}{\mathrm{A} + \mathrm{C}_{\mathrm{back}}} \tag{5}$$

where $\mathrm{C}_{\mathrm{back}}$ was the background intensity relative to final VLP intensity and $\mathrm{P}/S_{\mathrm{Background}}$ was the ratio of fluorescence when puncta $A = 0$, that is $\mathrm{P}/S_{\mathrm{Background}} = \mathrm{P}/S_{\mathrm{VLP}}(0, \rho(0))$.

We found an angle $\beta = 45°$, $\mathrm{C}_{\mathrm{back}} = 0.45$, and $d = 1$ approximately replicated the observed results (*Figure 4A and F*), reproducing the $t_{1/2\,\mathrm{of}\,\mathrm{P+2S}} - t_{1/2\,\mathrm{of}\,\mathrm{P/S}}$ of 0.16 (normalized time) (*Figure 4B*). In *Figure 4A* an exponential fit to P+2S was assumed in order to directly equate the VLP area to the predicted P/S. Similar results were obtained by using a uniform distribution of dipoles over an angular range, such as $\beta$ between 0° to 68.5° or 20° to 63.5°. Thus, our observation is consistent with

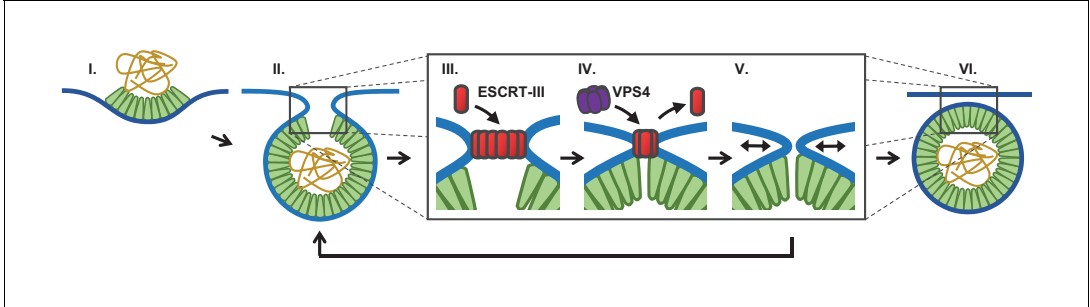

**Figure 5.** Proposed temporal model of ESCRT-III-mediated scission of HIV from cell plasma membrane. (I) The viral particle structure changes throughout accumulation of Gag until (II) a spherical topology prevents incorporation of additional Gags. (III) ESCRT-IIIs (examples: CHMP2, CHMP4) are recruited to the neck and polymerize, with (IV) removal by VPS4 resulting in constriction of the neck. (V) After removal of all ESCRT-IIIs the narrow neck undergoes spontaneous fission, (VI) freeing the virus. If membrane fission does not occur a new round of ESCRT-III recruitment is required (V→ II).
DOI: https://doi.org/10.7554/eLife.36221.021

formation of a spherical bud throughout the recruitment and multimerization of Gag (*Woodward et al., 2015*; *Carlson et al., 2008*), and is independent of the presence of early acting factors like ESCRT-I/TSG101 or ALIX (*Figure 4—figure supplement 2*). The bending also occurs many minutes before the recruitment of ESCRT-IIIs.

## Discussion

In retroviral assembly, the late domains of Gag are believed to indirectly recruit ESCRT-IIIs which then polymerize into ring or spiral structures at the bud neck to drive bending and scission of the neck membrane (*Cashikar et al., 2014*; *Fabrikant et al., 2009*; *Hanson et al., 2008*). However, our observations compel a new formulation of the role of ESCRTs in bending and scission. First, the initial bending of the membrane from a flat sheet to a spherical bud occurs during Gag assembly and does not require factors recruited by the late domains like ESCRT-1/TSG101 or ALIX. This bending also occurs prior to the arrival of ESCRT-IIIs and thus ESCRT-IIIs are also not required to initiate curvature. Instead, this process may be encouraged by Gag multimerization (*Briggs et al., 2009*; *Wright et al., 2007*). Second, the ESCRT-IIIs are only recruited after the accumulation of Gag is complete, indicating that Gag is not sufficient for recruitment. Third, the ESCRT-III/VPS4 are only recruited for tens of seconds and can no longer be detected at the moment of scission. While the ESCRTs and VPS4 cannot be detected in the tens of seconds prior to scission, it does not eliminate the possibility that a subpopulation of fewer than 20% of the ESCRT-III/VPS4, which are not detectable in the background fluctuations, stay around for a longer period of time (further discussed in Materials and methods).

Factors beyond the late domains which might assist ESCRT-III recruitment include activation through ubiquitination or phosphorylation of Gag or one of the early ESCRTs, such as ESCRT-II (*Meng et al., 2015*) or ESCRT-I/TSG101, which is co-recruited along with Gag (*Bleck et al., 2014*; *Jouvenet et al., 2011*). Additionally, ESCRT-III recruitment might be facilitated by high curvature in the neck or specific lipids recruited to these regions (*Lee et al., 2015*). These conditions may also facilitate recruitment of ESCRT-IIIs even in the absence of ESCRT-I or ESCRT-II, although likely at lower rates, potentially accounting for the much slower viral particle release rates (*Bendjennat and Saffarian, 2016*; *Meng et al., 2015*). The observation that the bulk of measured curvature of the nascent virion occurs prior to the recruitment of ESCRT-IIIs does not rule out any role for ESCRTs in membrane curvature. Indeed, the transient recruitment of ESCRT-IIIs may further constrict the neck linking the virion to the cell to prepare it for scission. The ESCRT-IIIs might facilitate scission by polymerizing into a spiral structure to constrict the neck ('polymerization constriction') (*Cashikar et al., 2014*; *Wollert et al., 2009*) or into a ring which when removed constricts the neck ('purse string constriction', *Figure 5*) (*Saksena et al., 2009*). The VPS4 may remove the ESCRT-IIIs potentially acting as a unfoldase (*Yang et al., 2015*). VPS4 may also remodel the ESCRT-IIIs throughout polymerization, potentially rearranging or tightening the structure (*Cashikar et al., 2014*). Remodeling is

consistent with our observation that recruitment of VPS4 was virtually contemporaneous with the ESCRT-III, with a slight 5 s lag in HEK293T cells and 10 s in HeLa cells (*Bleck et al., 2014*).

We suggest that fission can only occur when the neck is narrow and after the ESCRT-IIIs are removed (*Figure 5*). What is driving the scission event if ESCRT-IIIs are gone? When the neck is sufficiently narrow (a few nm) fission may be a spontaneous event, possibly through a hemifission intermediate (*Fabrikant et al., 2009*; *Kozlovsky and Kozlov, 2003*; *Liu et al., 2006*). Narrowing of the neck by ESCRT-IIIs may allow additional Gag molecules adjacent to the neck to oligomerize, thereby keeping the constricted structure stable while the ESCRTs are displaced. However, cryo-EM images indicate a gap in the Gag lattice might be present at the site of the neck (*Carlson et al., 2008*). Alternatively, constriction might encourage exchange or modification of phospholipids with shapes and charges that help to retain a narrow neck following ESCRT-III disappearance. The ESCRT-IIIs are dependent on phospholipids with negative charges. This could explain the greater than three fold increased scission rate at a lower pH (10% $pCO_2$), which would raise the proton concentration to that of the $pK_a$ (6.5 and 7.7) (*van Paridon et al., 1986*), thereby reducing the surface charges (*Figure 2F*). If scission does not occur sufficiently soon after removal of ESCRT-IIIs, a new round of ESCRT-III recruitment and assembly is required to achieve scission (*Figure 3*).

# Materials and methods

**Key resources table**

| Reagent type (species) or resource | Designation | Source or reference | Identifiers | Additional information |
|---|---|---|---|---|
| Cell line (*H. sapiens*) | HEK-293T | Bieniasz Lab | | |
| Cell line (*H. sapiens*) | HeLa | ATCC | | |
| Transfected construct (HIV-1, *A. victoria*) | Gag-mEGFP | *Bleck et al. (2014)* | | |
| Transfected construct (HIV, *A. macrodactyla*) | Gag-mTagBFP | *Bleck et al. (2014)* | | |
| Transfected construct (*A. victoria, H. sapiens*) | pLNCX2-mEGFP-VPS4A | *Bleck et al. (2014)* | | |
| Transfected construct (*A. marginale, H. sapiens*) | pLNCX2-mCherry-CHMP4B | *Bleck et al. (2014)* | | |
| Transfected construct (HIV) | GagΔp6 | *Bleck et al. (2014)* | | |
| Transfected construct (HIV) | pCR3.1/Syn-Gag | *Jouvenet et al. (2008)* | | |
| Transfected construct (HIV, *A. victoria*) | pCR3.1/Syn-Gag-pHluorin | *Jouvenet et al. (2008)* | | |
| Transfected construct (*A. marginale, H. sapiens*) | pLNCX2-mCherry-CHMP2A | This study | | CHMP4B in pLNCX2-mCherry-CHMP4B was replaced by CHMP2A from HEK293T cDNA library |
| Transfected construct (*A. marginale, H. sapiens*) | pLNCX2-mCherry-CHMP2B | This study | | CHMP4B in pLNCX2-mCherry-CHMP4B was replaced by CHMP2B from HEK293T cDNA library |
| Transfected construct (*A. marginale, H. sapiens*) | pLNCX2-mCherry-CHMP2A-siRNAres | This study | | added six silent point mutations to pLNCX2-mCherry-CHMP2A using site directed mutagenesis |
| Transfected construct (*A. marginale, H. sapiens*) | pLNCX2-mCherry-CHMP2B-siRNAres | This study | | added six silent point mutations to pLNCX2-mCherry-CHMP2B using site directed mutagenesis |
| Transfected construct (*A. marginale, H. sapiens*) | pLNCX2-mEGFP-CHMP2A | This study | | replaced mCherry in pLNCX2-mCherry-CHMP2A with EGFP (pEGFP-N1, Clontech) |
| Transfected construct (*A. marginale, H. sapiens*) | pLNCX2-mEGFP-CHMP2B | This study | | replaced mCherry in pLNCX2-mCherry-CHMP2B with EGFP (pEGFP-N1, Clontech) |

*Continued on next page*

*Continued*

| Reagent type (species) or resource | Designation | Source or reference | Identifiers | Additional information |
|---|---|---|---|---|
| Transfected construct (*A. marginale, H. sapiens*) | pLNCX2-mCherry-VPS4A | This study | | replaced mEGFP in pLNCX2-mCherry-VPS4A (*Bleck et al., 2014*) with mCherry (pmCherry- N1, Clontech) |
| Transfected construct (HIV, *A. victoria*) | Gag-Δp6-mEGFP | This study | | p6 domain removed from Gag-mEGFP using site directed mutagensis |
| Transfected construct (HIV, *A. victoria*) | Gag-MA-sf3 | This study | | sf3 GFP (*Pédelacq et al., 2006*) inserted in pCR3.1/Syn-Gag |
| Transfected construct (HIV, *A. victoria*) | Gag-MA-sf11 | This study | | sf11 GFP (*Pédelacq et al., 2006*) inserted in pCR3.1/Syn-Gag |
| Sequence-based reagent | CHMP2A DsiRNA | Integrated DNA Technologies | 5'-AAGAUGAAGAGGAG AGUGAUGCUdGdT-3' | |
| Sequence-based reagent | CHMP2A DsiRNA (reverse) | Integrated DNA Technologies | 5'-ACAGCAUCACUCUC CUCUUCAUCUUCC-3' | |
| Sequence-based reagent | CHMP2B DsiRNA | Integrated DNA Technologies | 5'-GGAACAGAAUCGAG AGUUACGAGdGdT-3' | |
| Sequence-based reagent | CHMP2B DsiRNA (reverse) | Integrated DNA Technologies | 5'-ACCUCGUAACUCUC GAUUCUGUUCCUU-3' | |
| Chemical compound, drug | DMEM | Thermo Fisher Scientific | #11965 | |
| Chemical compound, drug | FBS | MilliporeSigma | #F4135 | |
| Chemical compound, drug | HEPES | MilliporeSigma | #H3375 | |
| Software, algorithm | Metamorph | Molecular Devices | version 7.8.10 | |
| Software, algorithm | ImageJ | *Schneider et al. (2012)* | version Fiji | |
| Software, algorithm | LabView | National Instruments | version 2013 | |
| Software, algorithm | Microscope-Control | *Johnson, 2018c*) | | LabView Code |
| Software, algorithm | Average-puncta-center | *Johnson, 2018a*) | | LabView Code |
| Software, algorithm | Puncta-Fit | *Johnson, 2018d*) | | LabView Code |
| Software, algorithm | CO2-switch-analysis | *Johnson, 2018b*) | | LabView Code |

## Plasmid construction

Plasmids Gag-mEGFP, Gag-mTagBFP, pLNCX2-mEGFP-VPS4A and pLNCX2-mCherry-CHMP4B, Gag-Δp6 were described in *Bleck et al. (2014)*, and the plasmids pCR3.1/Syn-Gag and pCR3.1/Syn-Gag-pHluorin were described in *Jouvenet et al. (2008)*. The Gag in this study was based on the sequence from HIV-1 clone HXB2 (*Kotsopoulou et al., 2000*).

pLNCX2-mCherry-CHMP2A-siRNAres and pLNCX2-mCherry-CHMP2B-siRNAres were generated as follows. CHMP2A and CHMP2B were PCR amplified (Platinum PCR SuperMix, Thermo Fisher) from a cDNA library created from HEK293T cells (made via Invitrogen SuperScript III CellDirect kit #46–6320). PCR primers for CHMP2A (NM_014453.3) were 5'-GCGCTCCGGACTCAGA TCCCCGGAATTCATGGACCTATTGTTCGGGCG-3' and 5'-GCGCCTCGAGTCAGTCCCTCCGCAGG TTCT-3' and primers for CHMP2B (NM_014043.3) were 5'-GCGCTCCGGACTCAGATCCCCGGAA TTCATGGCGTCCCTCTTCAAGAA-3' and 5'-GCGCCTCGAGCTAATCTACTCCTAAAGCCT-3'. After PCR amplification, the fragments were digested with XhoI and BspEI (New England BioLabs, Ipswich, MA) and ligated into the plasmid pLNCX2-mCherry-CHMP4B (which was first digested with the same restriction enzymes to remove CHMP4B) using T4 DNA Ligase (New England BioLabs) yielding the plasmids pLNCX2-mCherry-CHMP2A and pLNCX2-mCherry-CHMP2B. Six silent coding mutations were then incorporated into these plasmids (QuickChange Lightning Site-Directed Mutagenesis Kit, Agilent, Santa Clara, CA) to make them insensitive to siRNA knockdown targeted at the cellular CHMP2A and CHMP2B RNA. The primers for the CHMP2A site-directed mutagenesis were 5'-ACC TGGGACACCACAGCATCGCTTTCTTCCTCGTCCTCCTCATCACCCATGGCATC-3' and 5'-GA TGCCATGGGTGATGAGGAGGACGAGGAAGAAAGCGATGCTGTGGTGTCCCAGGT-3' and for

CHMP2B were 5'-AACCGTGGATGATGTAATAAAGGAGCAAAACCGTGAATTACGAGGTACACA-GAGGGCTAT-3' and 5'-ATAGCCCTCTGTGTACCTCGTAATTCACGGTTTTGCTCCTTTATTACATCA TCCACGGTT-3'. The siRNA sequence for CHMP2A knockdown was 5'-rArArGrArUrGrArArGrArGr-GrArGrArGrUrGrAdTdT-3' (begins at position 464 in NM_014453.2) and for CHMP2B was 5'-rGrGrArArCrArGrArArUrCrGrArGrArGrUrUrAdTdT-3' (begins at position 45 in NM_014043.3) (Morita et al., 2011). The forward and reverse DsiRNA for CHMP2A knockdown was 5'-rArArGrAr-UrGrArArGrArGrGrArGrArGrUrGrArUrGrCrUdGdT-3' and 5'-rArCrArGrCrArUrCrArCrUrCrUrCrCrUr-CrUrUrCrArUrCrUrUrC-3'. The forward and reverse DsiRNA for CHMP2B knockdown was 5'-rGrArArCrArGrArArUrCrGrArGrArGrUrUrArCrGrArGdGdT-3' and 5'-rArCrCrUrCrGrUrArArCrUr-CrUrCrGrArUrUrCrUrGrUrUrCrCrUrU-3'. All ssDNA and ssRNA oligos were purchased from Integrated DNA Technologies (Coralville, IA) and manufacturers' protocols were used for all preparations.

pLNCX2-mEGFP-CHMP2A and pLNCX2-mEGFP-CHMP2B were generated by replacing mCherry in pLNCX-mCherry-CHMP2A and pLNCX-mCherry-CHMP2B with monomeric variant (A206K) of mEGFP from pEGFP-N1 (Clontech/Takara Bio USA, Mountain View, CA). Restriction enzymes AgeI and BspEI (New England BioLabs) were used to digest the backbone and fragments, and these fragments were then ligated into the backbones with T4 DNA ligase. The clonal HeLa cell lines were generated from these plasmids using a previously described protocol (Bleck et al., 2014). pLNCX2-mCherry-VPS4A was generated by replacing mEGFP in pLNCX2-mCherry-VPS4A (Bleck et al., 2014) with mCherry. mCherry was PCR amplified from pmCherry-N1 (Clontech) using In-Fusion recombination primers 5'-CTCTAGCGCTACCGGTCGCCACCATGGTGAGCAAGGGC-3' and 5'-CTCTAGCGC TACCGGTCGCCACCATGGTGAGCAAGGGC-3'. pLNCX2-mEGFP-VPS4A was digested with AgeI and BspEI and the fragment containing VPS4A was gel purified (Thermo Fisher PureLink Quick Gel Extraction Kit). The PCR product was then inserted into purified backbone using In-Fusion HD Cloning Kit (Clontech) according to manufacturer's instructions.

Gag-Δp6-mEGFP was generated by deleting p6 and SP2 from Gag-mEGFP via site-directed mutagenesis (QuickChange Lightning Site-Directed Mutagenesis Kit). The following primers were used for the deletion 5'- TACTGAGAGACAGGCTAATTCGGATCCACCGGT-3'; 5'- ACCGGTGGA TCCGAATTAGCCTGTCTCTCAGTA −3'. Gag-MA-sf3 and Gag-MA-sf11 were generated by inserting the circularly permuted superfolder GFPs (provided by Jeffrey Waldo lab) (Pédelacq et al., 2006) into a variant of pCR3.1/Syn-Gag that has an EcoRV restriction enzyme site near the carboxy terminal of MA (Asn-Gln-Val-Ser modified to Asn-Gln-**Asp-Ile**-Val-Ser). Circularly permuted version 3 was PCR amplified with recombination In-Fusion primers 5'-CACAGCAACCAGGATGGCAG-CAGCCATCATCATC-3' and 5'-GTTCTGGCTGACGATGGTACCTCCAGTAGTGCAAATAA-3'. The primers for circularly permuted version 11 were 5'-CACAGCAACCAGGATGGCAGCAGCCATCA TCATC-3' and 5'-GTTCTGGCTGACGATGGTACCATCTTCAATGTTGTGG-3'. After digesting pCR3.1/Syn-Gag-EcoRV with EcoRV (New England BioLabs) the In-Fusion HD kit was used to insert PCR fragments into the Syn-Gag backbone.

## Sample preparation

With the exception of 120 s $pCO_2$ switching data, all imagings were conducted in HEK293T cells grown in DMEM (#11965, Thermo Fisher Scientific, Waltham, MA) with 10% FBS (#F4135, Millipore-Sigma, St. Louis, MO). HEK293T cells were gift from P. Bieniasz Laboratory and were not further authenticated but tested negative for mycoplasma contamination. 120 s $pCO_2$ switching experiments imaged in HeLa cells which were grown in the same growth medium. HeLa cells were from ATCC and were not authenticated or tested for mycoplasma contamination. For polarization excitation experiments, cells were grown on 35-mm glass bottom dishes (#P35G-1.0–20 C, MatTek, Ashland, MA) coated with fibronectin (#33010, Thermo Fisher) by incubating the dish with 10 μg/ml fibronectin in PBS for 1 hr. For $pCO_2$ switching experiments, perfusion slides (#80186 μ-Slide, Ibidi, Martinsried, Germany) were incubated with 60 μg/ml fibronectin in PBS for 1 hr before plating cells. For all experiments, cells at ∼ 75% confluency were transfected with expression plasmids ∼3.5 hr prior to beginning to image. For transfection 8 μl of Lipofectamine2000 (#11668, Thermo Fisher) was incubated for 5 min in 250 μl of Opti-MEM I (#31985, Thermo Fisher) and 2000 μg of DNA was incubated for 5 min in 250 μl of Opti-MEM I. Both solutions were then mixed and incubated for 20 min before adding to adhered cells on MatTek dish in 2000 μl of DMEM. The same procedure was used for cells in the flow slide, but the transfection mixture was added to 2000 μl of DMEM and

then 1000 μl was perfused through the chamber. At least four cells were used for each experimental condition in order to account for potential variability between cells. Both CHMP2A and CHMP2B were knocked down with siRNA for mCherry-CHMP2A and mCherry-CHMP2B experiments. siRNA transfections were conducted 48 hr prior to DNA plasmid transfection using Lipofectamine RNAi-MAX (following manufacturer's instructions; #13778, Thermo Fisher). A second round of siRNA transfection was performed at the time of DNA transfection.

The open-reading frame was verified via sequencing for all plasmids and the following DNA ratios were used for transfections: Gag:Gag-mEGFP (4:1), Gag:Gag-MA-sf3 (4:1), Gag:Gag-MA-sf3 (4:1), Gag-Δp6:Gag-Δp6-mEGFP (4:1), Gag:Gag-pHluorin:mCherry-VPS4A (12:3:5), Gag:Gag-pHluorin:mCherry-CHMP4B (12:3:5), Gag:Gag-pHluorin:mCherry-CHMP2A-siRNAres (4:1:5), Gag:Gag-pHluorin:mCherry-CHMP2B-siRNAres (4:1:5), Gag:Gag-TagBFP:mEGFP-VPS4A:mCherry-CHMP4B (24:6:5:5). Note: The Gag to tagged Gag ratio was 4:1 in all experiments.

## $CO_2$ modulation system

Gas from compressed cylinders was bubbled into two imaging media reservoirs (140 ml open piston Monoject syringe, Medtronic, Minneapolis, MN) partially filled with cell imaging media (10 mM HEPES, 9.7 g/L of Hanks BBS (MilliporeSigma), and NaOH to adjust the pH to 7.4) with 1% FBS. One reservoir was equilibrated with compressed air (labeled 0% in these experiments but actually contained 0.04% $CO_2$), and the other reservoir was equilibrated with 10% $CO_2$ (balanced with air) (*Figure 1—figure supplement 1*). The reservoirs were mounted to a flow perfusion system (ValveBank II, AutoMate Scientific, Berkeley, CA), which enabled automated selection of desired media via solenoid valves under the reservoirs. Tygon tubes (R-3603) from each valve carried the media to a fluid combiner just prior to the perfusion slide chamber containing adhered cells. After the flow chamber, a single tube carried the discharge media to a collection container. This container was placed below the height of equilibration reservoirs so that fluid flow was driven by gravity. The flow rate (~3 ml/min) was controlled with a clamp regulator attached to the discharge tube. At this flow rate, the response time of the fluorophores to a $pCO_2$ change, characterized in terms of an exponential decay constant, was ~7 s. A peristaltic pump (MS-Reglo, Ismatec/Cole-Parmer, Wertheim, Germany) then passed the media from the collection reservoir back to the equilibration reservoirs so that the media could be recycled. The microscope and entire imaging media flow system were enclosed in a temperature control box held at 37°C. Valve regulation, camera trigger and laser excitation were all controlled via custom software (https://github.com/SimonLab-RU/Microscope-Control; copy archived at https://github.com/elifesciences-publications/Microscope-Control) written in LabView (National Instruments, Austin, TX).

## Imaging with $pCO_2$ switching

For experiments with $pCO_2$ switching every 10 s the media was continuously flowed through the imaging chamber, with the reservoir supplying the media being switched every 10 s. Two images were collected every 2.5 s (4.0 s for CHMP4B experiments) with 488 nm excitation for pHluorin (100 ms exposure with power between 1 and 5 mW), followed by 594 nm excitation for mCherry (100 ms exposure with laser power between 5 and 20 mW; 100 mW DPSS laser, Cobolt, Solna, Sweden). A multipass emission filter (zet405/488/594 m, Chroma Technology, Bellows Falls, VT) enabled rapid sequential wavelength imaging. For experiments with $pCO_2$ switching every 120 s the desired media was only applied to the chamber for 10 s, followed by no flow for 110 s. During this time sequential 488 nm (100 ms, 1 mW) and 594 nm (100 ms, 1 mW) excitation images were captured every 10 s. All experiments were conducted with 100 Hz azimuthal scanning TIRF microscopy illumination.

## Temporal determination of scission

The cytosolic pH was oscillated by switching the $pCO_2$ from 0% to 10% (for an average of 5%) every 10 s or every 120 s. The cytosolic carbonic anhydrase ensures that the pH in the cytosol closely tracks the $pCO_2$ (*Simon et al., 1994*). At scission, the luminal pH of the VLP or virion is no longer continuous with the cytosol and no longer tracks the cytosolic pH. Individual VLPs were identified from the Gag-pHluorin images using Metamorph (version 7.8.10, Molecular Devices, Sunnyvale, CA) and the peak pixel amplitude in a 15 pixel (975 nm) diameter regions of interests centered on individual VLPs was found for all frames (*Figure 1*, *Figure 1—figure supplement 2*).

For switching every 10 s, we used a custom LabView software lock-in amplifier to find changes in VLP sensitivity to $pCO_2$ modulation (https://github.com/SimonLab-RU/CO2-switch-analysis; copy archived at https://github.com/elifesciences-publications/CO2-switch-analysis). The data were high-pass filtered (2-pole Butterworth with 0.01 Hz cutoff) to bias the $pCO_2$-dependent pHluorin intensity fluctuations (with a period of 20 s) around 0. This signal was then multiplied by an in-phase sine wave with the same period and a moving average was calculated over a period of 60 s. A significant change in signal indicated a change in sensitivity to $pCO_2$ modulation. At many of the identified VLPs, a clear transition from a high plateau to a low plateau in the lock-in signal was observed (*Figure 1—figure supplement 2*). The moment of scission was classified as the halfway amplitude between the two plateaus, with half of the moving window containing pre-scission data and the other half containing post-scission data (*Figure 1—figure supplement 2A*). In the same regions of interest, the average mCherry signal (tagged to VPS4A, CHMP4B, CHMP2A, or CHMP2B) was analyzed and peaks were identified in which there was a clear increase followed by decrease in fluorescence intensity (*Figure 1A,C,E,G* and *Figure 1—figure supplement 2*). The appearance time was identified when a signal was first observed above cellular background, and the disappearance time when the signal dropped to cellular background. Appearance and disappearance relative to scission were then found by comparing these times to the scission time (*Figure 1B,D,F,H*). Based on the observed distribution of appearance and disappearance times relative to scission, with an average of roughly a minute, we excluded data in which scission and ESCRT-III/VPS4A recruitment were more than 4 min apart. Excluded data was attribute to a failed or uncorrelated scission event. ~25 traces under each condition were collected to gain understanding of the distribution of events.

The average signal-to-noise was approximately 7:1 (peak signal:S.D.) across all ESCRT-III/VPS4A measurements, and ESCRT-III/VPS4A recruitment was estimated to be detectable when the sustained signal deviated from the mean by ~ 1–2 standard deviations. Based on this deviation, it is estimated that ESCRT-III/VPS4A recruitment below ~20% of peak recruitment would be undetected. We estimate that there are between 10 and 100 ESCRTs in a complex at peak signal, with the possibility of < 10 ESCRTS undetectable in the background noise. Assuming a peak signal (100 ESCRTs) decreases exponentially into the noise after 10 s (10 ESCRTS), we estimate only ~1 ESCRTs would remain after another 10 s. This time is comparable to the ~20 s measured between peak disappearance and scission. If the disappearance is faster than exponential, for example linear, the ESCRTs will be gone even sooner. This interpolation of ESCRT-IIIs or VPS4A disappearance profiles into the noise indicates the ESCRTs predominantly leave the budding site prior to scission (*Figure 1—figure supplement 2*). For a significant number (>1) of either ESCRT-IIIs or VPS4A to be around following scission there would need to be a second, smaller population that follows much slower disappearance kinetics.

For $pCO_2$ switching every 120 s for each tracing we determined the first transition of $pCO_2$ for which the fluorescence had decreased sensitivity, indicating protons were no long freely flowing between cytosol and lumen of the VLP. We assumed that scission must have occurred during the previous plateau of $pCO_2$ (*Figure 2A–E*). For example, if scission occurred during the 10% $pCO_2$ cycle (low fluorescence) then after the next transition to 0% $pCO_2$, and all subsequent transitions, the fluorescence would be closer to that of the 10% (low fluorescence state) than the 0% $pCO_2$ (high fluorescence) state. Conversely, if scission occurred during the 0% (high fluorescence), future 10% $pCO_2$ plateaus would be closer to the high fluorescence state. Occasionally traces appeared to be trapped in an intermediate pH state, which we attribute to scission occurring during the transition between $pCO_2$ states. Individual VLP traces were categorized into being trapped in a high, middle, or low pH state by comparing the Gag-pHLuorin signal before and after the VLP became insensitive to $pCO_2$ switching (*Figure 2F*).

## Simultaneous VPS4A and CHMP4B imaging

Gag-mTagBFP, mEGFP-VPS4A and mCherry-CHMP4B were imaged with TIRF with sequential illumination of 594 nm (100 ms, 10–20 mW), 488 nm (100 ms, 4–10 mW) and 405 nm (100 ms, 2–4 mW; 120 mW LuxX diode laser, Omicron). mEGFP-VPS4A and mCherry-CHMP4B images were acquired every 5 s and Gag-TagBFP images were acquired every 60 s. VPS4A and CHMP4B relative appearance time were conducted using the same method as described previously (*Figure 1—figure supplement 4*) (*Bleck et al., 2014*).

## mEGFP-CHMP2A/B knockdown imaging

HeLa cell lines stably expressing either mEGFP-CHMP2A or mEGFP-CHMP2B and transfected with the respective siRNA (48 hr in advance) were imaged on an inverted microscope (IX-70, Olympus, Shinjuku, Japan) with epi-illumination via a Xenon lamp and transmitted bright-field illumination (*Figure 1—figure supplement 3*).

## Polarized excitation imaging

Images were collected on an inverted microscope (IX-81, Olympus) with a custom built through-the-objective (100X UAPON 1.49 NA, Olympus) polarized TIRF microscopy illuminator (*Johnson et al., 2014*). Throughout imaging the excitation TIR light was azimuthally scanned at 200 Hz with mirror galvanometers (Nutfield Technology, Hudson, NH) in order to reduce spatial illumination nonuniformities. A multiband polychroic (zt405/488/594/647rpc 2 mm substrate, Chroma) was positioned between the galvanometers and objective in order to isolate the excitation light from the emitted light. Light from a 488 nm laser (100 mW LuxX diode laser, Omicron, Rodgau-Dudenhofen, Germany) was modulated between being polarized perpendicular ($\hat{p}$) or parallel ($\hat{s}$) to the glass surface by passing the light through an electro-optic modulator (EOM) (Conoptics, Danbury, CT) and quarter-wave plate prior to the galvanometer scan-head. The polarization generated by the EOM was modulated in sync with the galvanometers such that during scanning a $\hat{p}$ or $\hat{s}$ state was maintained at all azimuthal positions. The scanning polar angle was selected such that the excitation light was just beyond the TIR critical angle, minimizing $\hat{s}$ polarized light contaminating $\hat{p}$ excitation (*Johnson et al., 2014*).

A combined $\hat{p}/\hat{s}$ ratio image was collected every 30 s. To generate this ratio image a sequential series of 10 $\hat{p}$ and $\hat{s}$ images were collected, divided (after subtracting camera offset), and then averaged. Each $\hat{p}$ or $\hat{s}$ image had an exposure of 5 ms (laser power between 25 and 50 mW), with a new image collected every 15 ms. Thus, in 30 ms, a single $\hat{p}/\hat{s}$ image was generated, and then 10 of these images (over a 300 ms duration) were averaged (ImageJ) (*Schneider et al., 2012*) to create the combined ratio image. The short period was utilized in order to minimize artifacts in the $\hat{p}/\hat{s}$ ratio image from VLP or cell movement. The galvonometers, EOM, camera shutter, and laser shutters were all driven/triggered by a multifunction data acquisition board (PCIe-6323, National Instruments) and controlled from custom written software in LabView (https://github.com/SimonLab-RU/Microscope-Control; copy archived at https://github.com/elifesciences-publications/Microscope-Control). Images were streamed from a CMOS camera (Flash-4.0, Hamamatsu, Hamamatsu City, Japan) to a workstation (T7500, Dell, Round Rock, TX) running image acquisition software (Metamorph). A single band emission filter (ET525/50 m, Chroma) was used to isolate fluorophore emission. In order to characterize the amount of Gag in an assembling VLP, an average $\hat{p} + 2 \cdot \hat{s}$ image was also generated at each time point.

## Polarization analysis

Puncta were found in the $\hat{p} + 2 \cdot \hat{s}$ images which increased and then plateaued in amplitude. These puncta were then selected for orientation analysis with $\hat{p}/\hat{s}$. A 2D Gaussian was fit to these assembling puncta to find a frame by frame subpixel peak location (https://github.com/SimonLab-RU/Puncta-Fit; copy archived at https://github.com/elifesciences-publications/Puncta-Fit). Using bilinear interpolation, the $\hat{p}/\hat{s}$ and $\hat{p} + 2 \cdot \hat{s}$ images were resampled 10X in the horizontal and vertical directions (65 to 6.5 nm wide pixels). On these resampled images P/S and P+2S values were found by averaging the resampled pixel intensities that are within 100 nm of the peak fit locations (https://github.com/SimonLab-RU/Average-puncta-center; copy archived at https://github.com/elifesciences-publications/Average-puncta-center) (*Figure 4A*, *Figure 4—figure supplement 1*). For frames prior to the appearance of a puncta P/S and P+2S values were determined using the fit location of the first frame in which there was a puncta fit. For each tagged version of Gag, the average P/S value for background and assembled VLPs was calculated by finding an average P/S for each trace before assembly and after a plateau was reached, and then averaging across all traces (*Figure 4B*). A relative time to half growth for each VLP was determined by finding the halfway to assembly point (i.e. the point where the intensity is halfway between the intensity at assembly beginning and plateau) and then finding the normalized time at this point relative to the time assembly began and appeared to reach a plateau. This normalized time was between 0 and 1. A normalized time to

halfway drop in P/S was also found (normalized to the same time scale) and these values were subtracted to find: $t_{1/2 \text{ of } P+2S} - t_{1/2 \text{ of } P/S}$ (normalized). All fluorophore combinations were included in the P/S histogram (*Figure 4C*) since all combinations had similar P/S characteristics.

## Acknowledgements

We thank M Will and M A Lockard for assisting with plasmid preparations. We also thank M S Itano, M D Tomasini, K Bredbenner and P Bieniasz for discussion during manuscript preparation. This work was supported by NIH grants 1P50GM103297 and R01 GM11958 to SMS.

## Additional information

### Funding

| Funder | Grant reference number | Author |
| --- | --- | --- |
| National Institutes of Health | 5R01GM119585 | Daniel S Johnson<br>Marina Bleck<br>Sanford M Simon |
| National Institutes of Health | 2U54GM103297 | Daniel S Johnson<br>Marina Bleck<br>Sanford M Simon |

The funders had no role in study design, data collection and interpretation, or the decision to submit the work for publication.

### Author contributions

Daniel S Johnson, Conceptualization, Data curation, Software, Formal analysis, Validation, Investigation, Visualization, Methodology, Writing—original draft, Project administration, Writing—review and editing, Designed experiments, Prepared samples, Collected and analyzed data; Marina Bleck, Conceptualization, Data curation, Formal analysis, Investigation, Visualization, Writing—original draft, Writing—review and editing, Designed experiments, Prepared samples; Sanford M Simon, Conceptualization, Formal analysis, Supervision, Funding acquisition, Investigation, Methodology, Writing—original draft, Project administration, Writing—review and editing, Designed experiments

### Author ORCIDs

Daniel S Johnson (iD) http://orcid.org/0000-0001-8906-0509
Marina Bleck (iD) http://orcid.org/0000-0003-4175-8363
Sanford M Simon (iD) http://orcid.org/0000-0002-8615-4224

### Decision letter and Author response

Decision letter https://doi.org/10.7554/eLife.36221.025
Author response https://doi.org/10.7554/eLife.36221.026

## Additional files

### Supplementary files

• Transparent reporting form
DOI: https://doi.org/10.7554/eLife.36221.022

### Data availability

Data generated or analyzed during this study are included in the manuscript and supporting files.

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
