## [Decision Letter]

Thank you for submitting your article "Timing of ESCRT-III protein recruitment and membrane scission during HIV-1 assembly" for consideration by *eLife*. Your article has been reviewed by 3 peer reviewers, one of whom is a member of our Board of Reviewing Editors, and the evaluation has been overseen Wendy Garrett as the Senior Editor. The following individuals involved in review of your submission have agreed to reveal their identity: Stuart Neil (Reviewer #2); Andrew Lever (Reviewer #3).

The reviewers have discussed the reviews with one another and the Reviewing Editor has drafted this decision to help you prepare a revised submission.

Our reviewers had several suggestions for improvement. We include here their reviews and some ranking of the importance of responses to their comments.

Editorial comments on the reviews below:

Re: Review #1:

No edits requested.

Reviewer #1

The mode of action of the ESCRT proteins in narrowing and pinching off a membranous neck has been a puzzle for many years. Using TIRF imaging of budding virus, this paper provides much new information about the process, mainly concerning the timing of the ESCRT presence at the buds. The data show the arrival of various components late in the game, and then (significantly) their departure right before closure. There is nice quantification of the ESCRTs. The strength of the paper is the detailed measurement of the timing. The findings explain such issues as the absence of the ESCRTs in the final virions and limit the models for the structures used for the narrowing. Various manipulations of the system reveal some parameters of the reaction. Basically, this is cutting-edge imaging that is pushed to give great kinetic and stoichiometric information about ESCRT functions. The paper represents a significant advance in the field.

Re: Comments on Review #2:

This review is strongly positive.

Adding some discussion of the number of waves of recruitment that are typically seen would be very helpful. Highly encouraged. The reviewer is surprised by the inefficiency of this system, so some comments on that might be warranted. Whether the need for multiple tries would be true of wild-type virus might be raised.

Looking at L-domain-negative mutants could be interesting, but the extent of L-domain-negative budding might be too low to measure, or too aberrant to evaluate. This is probably of lower priority. Could be addressed by discussion.

Including movies would be attractive indeed. We could imagine it may be difficult to provide ones that convey the data in a format that is helpful to readers, but *eLife* is good at this. Try to add these.

The carbonic anhydrase measurements would be helpful if data are in hand.

Reviewer #2:

Having previously established TIRF microscopy techniques to analyse the budding of HIV-1 virions in real time, and the coordination of the recruitment of ESCRT proteins, the Simon lab now further develop these studies to the recruitment and disappearance of individual ESCRT-III components and VPS4 in the context of the actual membrane scission event itself. To visualize the moment of scission by live imaging, the authors have adapted their CO_2_-mediated manipulation of cytoplasmic pH and pHluorinGFP-fused Gag by imaging assembly and Gag recruitment under conditions where CO_2_ concentrations oscillate every 10s. Since the pHluorin in released virions becomes much less sensitive to these perturbations, this allows the authors to precisely define the moment of membrane scission in live cells – a really inventive solution to a key problem. By doing this, the authors show that ESCRT-III components and VPS4 are removed from the virion several seconds before the scission event, favouring a model whereby ESCRT-III constricts the neck of the virion to a point favourable to scission but must be removed by VPS4 to allow the reaction to take place rather than directly participating in it. If scission fails after their removal, subsequent rounds of ESCRT-III recruitment appear to occur to allow viral release. By varying the time of pH oscillation, the authors find that low cytoplasmic pH promotes scission. From this they suggest that protonation of phospholipid heads in the constricted neck may explain this, raising the possibility that it is proximity of the lipid heads that drive local destabilization resulting scission. Finally, using GFP variants embedded in Gag that only fluoresce under polarized excitation and therefore are restricted in an assembling particle, the authors model the membrane curvature of the budding virion, providing realtime evidence that Gag alone is the driving force of budding, and ESCRT-III recruitment occurs only when the spherical structure is formed.

Overall, this is a very interesting manuscript whose key strength is imaging of ESCRT function in relation to precise timing of scission in live cells using inventive and complex methodology. Despite this, the take home message is simple and important. I have some comments and queries below that the authors should consider addressing:

1) It is not clear from the manuscript how many waves of ESCRT-III recruitment happen for the average budding event. Do most scission events require only one? How many fail to progress to scission over the time frame despite multiple waves of ESCRT-III recruitment?

2) It is well established in artificial systems that Gag can induce membrane curvature, but the key novelty here is measuring it in real time in live cells. However, to definitively demonstrate ESCRT-independence, it would seem important to me to examine the membrane curvature of assembly of a Gag lacking the late domain in comparison – one should expect the same profile but subtle differences in timing might be interesting.

3) The authors should include representative videos of the events described.

4) Can the authors measure the relative concentration to carbonic anhydrase in purified virions vs the cytoplasm?

Re: Comments on Review #3:

This reviewer is also positive, with some suggested experiments. Some comments on his suggestions:

The issue of whether the readouts are revealing closure of the sphere, or pinching off of the membrane, is a real one, but it is not clear whether or not this can be fully determined experimentally (perhaps some aspects of this are known already?). At a minimum the issue should be directly discussed.

Looking at budding mutants (L-domain mutants again?) might be informative but might be hard to interpret, as noted above. It seems likely the authors have already looked at these, and if so, the issue can be addressed by discussion.

If there is a discrepancy with earlier work with respect to to CHMP4 recruitment, it should be addressed.

Comparing the effects on budding seen here, with effects on other host-mediated fusion effects (e.g. MVB fusion) would be helpful but might require setting up entire new assays. This would not seem absolutely required for acceptance.

The potential roles of Gag-Pol and viral RNA in the processes examined here are very interesting issues. I would include PR-mediated processing. But repeating all the work with and without these factors is a major effort, and it would probably be adequate to discuss the potential roles.

ESCRT-II is very likely involved and should be mentioned.

Reviewer #3:

The authors have used a number of interesting biophysical techniques to analyse the later stages of HIV budding and the temporal dynamics of recruitment of ESCRT-III components and membrane scission. Oscillating fluorescence of a pH dependent fluorophore was used to determine the point of membrane scission. I am curious to know whether they are actually measuring completion of the Gag sphere or actual membrane pinching off. In the original work of Gottlinger many of the particles that remained adherent to the cell appeared to have completed Gag sphere formation. Would it be useful to test out their oscillating pH system using such a budding mutant to determine whether they see slowing of oscillation caused by Capsid shell completion without particle release?

They previously showed ESCRT-III recruitment kinetics using a fluorophore based super resolution microscopy approach so some of the data here is more confirmatory rather than novel. The timing of CHMP4 recruitment and departure appears to be shorter than in their previously published work.

Acidification of the cytosol accelerates scission. My understanding is that the inner surface of the plasma membrane carries a generalized negative charge, so I am not sure how important the PIP_2_ components are in affecting scission. PIP_2_ acts as an important bridge binding the Matrix region of Gag to the inner plasma membrane but this occurs with all the Gag polyproteins from the first stages of assembly even before membrane bending and is not specific to those interactions involved at the budding neck. It would be of interest to see whether formation of vesicles triggered by ESCRT, such as in MVB formation, is similarly affected by pH changes to define how specific this is (or isn't) for virion budding and whether PIP_2_ is of specific importance.

The demonstration of repeated recruitment and loss of VPS4A and CHMP4B is interesting and novel. It begs the question as to what is the critical trigger for scission as it would appear that although they are important for the process they themselves are necessary but not sufficient. A significant caveat is that the experiments as far as I can tell are performed with Gag alone and there is evidence that the Gag/Pol polyprotein may influence budding significantly with a suggestion that cargo size affects ESCRT function (Bendjennat et al., 2016). Similarly, there is no genomic RNA, and this may affect virion assembly. Either of these may be involved in effecting the scission pathway and might be part of the definitive trigger, which would render repeated VPS4A/CHMP4B recruitment unnecessary.

I note that nowhere is the role of ESCRT-II mentioned. There is now clear biochemical and EM evidence that it is involved in HIV budding (Meng et al., 2015).

The membrane bending experiments are elegant although I am not mathematical enough to give an opinion on the calculations although the results are highly plausible.

---

## [Author Response]

Reviewer #2:[…] Overall, this is a very interesting manuscript whose key strength is imaging of ESCRT function in relation to precise timing of scission in live cells using inventive and complex methodology. Despite this the take home message is simple and important. I have some comments and queries below that the authors should consider addressing:1) It is not clear from the manuscript how many waves of ESCRT-III recruitment happen for the average budding event. Do most scission events require only one? How many fail to progress to scission over the time frame despite multiple waves of ESCRT-III recruitment?

A majority of our scission events occurred following the first recruitment of ESCRTs, but multiple rounds of ESCRT recruitment were also observed. In this work approximately 20% of our scission events consisted of more than one round of ESCRT recruitment. This is in rough agreement with our previous work in which we quantified the number of rounds of waves of recruitment (Jouvenet et al., 2011). In the previous work we did not simultaneously study scission, but in the current manuscript we demonstrate that scission only occurs following the last ESCRT recruitment. We have added a statement to the main text to clarify how often we observed multiple recruitment events. We rarely observed the recruitment of ESCRT-III without subsequent scission (~5%). However, we imaged cells for only a finite time and for those few events where we did not observe scission, we believe it occurred after we stopped imaging.

2) It is well established in artificial systems that Gag can induce membrane curvature, but the key novelty here is measuring it in real time in live cells. However, to definitively demonstrate ESCRT-independence, it would seem important to me to examine the membrane curvature of assembly of a Gag lacking the late domain in comparison – one should expect the same profile but subtle differences in timing might be interesting.

To address this question, we have repeated the Gag assembly experiments with either the intact Gag or with a deletion of the late domain (p6) of Gag, which contains sites for ESCRT recruitment. These were imaged and quantified using the same polarization-based excitation technique to monitor the orientation of the fluorophore (new Figure 4A). By measuring relative emission of GFP when excited with an electric field parallel (S-polarized) or perpendicular (P-polarized) to the glass we observed a similar change in the emission ratio (P-polarized/S-polarized) in the native Gag and Gag deleted of p6. Like wild type Gag assembly, the Gag missing p6 also had an initial P/S ratio of ~2 which then dropped to ~1.4 during the early parts of VLP formation (example trace added to Figure 4A; and additional examples shown in a new figure, Figure 4—figure supplement 2). We attribute this change to the VLP assembling into a spherical shell throughout assembly.

These results demonstrate that eliminating the late domain and the recruitment of early acting ESCRTs (like TSG101 or ALIX), have no effect that can be detected on the timing or extent of membrane bending. Thus, they are not required for VLPs to form the spherical shell structure, consistent with previous electron microscopy results in which Gag missing the late domain still formed spherical shells (Martin-Serrano and Bieniasz, 2003).

3) The authors should include representative videos of the events described.

We have now included representative movies for the following scenarios:

Video 1: Gag-pHluorin assembly (left) in whole cells with CO_2_ switching every 10s along with simultaneous recruitment of mCherry-CHMP4B (right).

Video 2: Example of individual puncta of Gag-pHluorin assembly with CO_2_ switching (left, every 10s) with simultaneous recruitment of mCherry-CHMP4B (right). CHMP4B was visible starting at 13:29 min.

Video 3: Gag-pHluorin assembly (top) in whole cells with CO_2_ switching every 10s along with simultaneous imaging of mCherry-VPS4A recruitment (right).

Video 4: Example of individual puncta of Gag-pHluorin assembly with CO_2_ switching (left, every 10s) with simultaneous recruitment of mCherry-VPS4A (right). VPS4A was visible starting at 20:49 min.

Video 5: Gag-pHluorin assembly (left) in whole cells with CO_2_ switching every 120s along with simultaneous imaging of mCherry-VPS4A recruitment (right).

Video 6: Example of individual puncta of Gag-pHluorin assembly with CO_2_ switching (left, every 120s) with simultaneous recruitment of mCherry-VPS4A (right). VPS4A was visible starting at 14:33 min.

Video 7: Gag-mEGFP assembly in whole cells with ratio image of perpendicular and parallel excitation (P/S) and total emission (P+2S).

Video 8: Example of individual puncta of Gag-mEGFP with P/S and P+2S images

We have added references and legends for these videos to the text.

4) Can the authors measure the relative concentration to carbonic anhydrase in purified virions vs the cytoplasm?

This is a good question, which has previously been addressed. Mass spectrometry studies of purified virions of HIV-1 that have probed for cellular proteins could not detect the presence of carbonic anhydrase (Ott, 2008). This is consistent with our observation that in isolated VLPs, modulating the pCO_2_ does not affect the pH in the VLPs. If there was significant carbonic anhydrase in isolated VLPs then CO_2_ modulation would result in pH (and thus fluorescence) changes – which we don’t observe (this work and Jouvenet, 2008).

Thus, the change in pH reported by the Gag-pHluorin in the forming bud is not the consequence of carbonic anhydrase in the bud. It is the result of a diffusive pathway in the neck that is sufficient size to allow the transit of protons. After scission, the connecting pathway is lost and protons in the cell cannot enter the nascent VLP. We have expanded the text to clarify our hypothesis on carbonic anhydrase in the VLPs.

Reviewer #3:The authors have used a number of interesting biophysical techniques to analyse the later stages of HIV budding and the temporal dynamics of recruitment of ESCRT-III components and membrane scission. Oscillating fluorescence of a pH dependent fluorophore was used to determine the point of membrane scission. I am curious to know whether they are actually measuring completion of the Gag sphere or actual membrane pinching off. In the original work of Gottlinger many of the particles that remained adherent to the cell appeared to have completed Gag sphere formation. Would it be useful to test out their oscillating pH system using such a budding mutant to determine whether they see slowing of oscillation caused by Capsid shell completion without particle release?

As described above, we believe that we are measuring scission of the membrane as opposed to completion of the Gag shell. Electron microscopy studies (Briggs et al., 2009; Carlson et al., 2008; Woodward et al., 2015; Wright et al., 2007) show that the immature Gag lattice is generally incomplete with openings in the Gag lattice at the junction with the mother cell, thus continued movement of protons back and forth. The rapid transition to a state where protons cannot move between the mother and bud indicates a loss of continuity thus scission. Based on our results carbonic anhydrase is not in the nascent VLP. Additionally, mass spectrometry of nascent virions looking at cellular proteins did not detect carbonic anhydrase in the virions (Ott, 2008). Thus, any CO_2_ that passes the membrane of the VLP can only be very slowly converted to bicarbonate and protons. We have previously demonstrated that with a late domain mutant, the Gag inside the nascent virion remains sensitive to the cytosolic pH (Jouvenet, 2008, Figure 3F).

They previously showed ESCRT-III recruitment kinetics using a fluorophore based super resolution microscopy approach so some of the data here is more confirmatory rather than novel. The timing of CHMP4 recruitment and departure appears to be shorter than in their previously published work.

First, one of the key novelties in this work is imaging the recruitment kinetics of ESCRT-III simultaneously with membrane bending or simultaneously with scission. This has not been done before by us, or anyone else.

Second, there is a difference in relative timing between recruitment of CHMP4B and VPS4A, but it is a difference not between previous work and this work, but a subtle difference that varies with the cell type. In both this work, which mainly examined HEK293T cells, and in our previous work on HeLa cells (Bleck et al., 2014), we monitored the time course of recruitment of CHMP4B simultaneously with recruitment of VPS4A. In both cell types, after ESCRT-III appeared (see red tracing in Figure 3 of the current manuscript), we detected VPS4A (see the green tracing). The ESCRT-III fluorescence increased, with the VPS4A following. The CHMP4B reached a maximum shortly before VPS4A, and then they both decreased.

In the HEK293T cells we observed that the lag to VPS4A reached its ½ max 5.1 seconds after CHMP4B and in the HeLa cells the lag was 9 seconds (Author response image 1 and Figure 1E from our previous paper, Bleck et al., 2014). From re-examining the histograms we are uncertain what, if any, the significance is of that delay. For this work we performed the majority of measurements in HEK293T cells because scission was observed to occur much more frequently, allowing us to study many more events. The difference in the lag between the HeLa and HEK293T is consistent between our previous work and this work (here we examined both). However, we did not examine the difference in the lag closely and therefore we are reticent to make any statements about its biochemical significance. For this publication we focused on the relative timing of the ESCRTs and VPS4 to the scission – where there is a much greater and consistent difference.

**Author response image 1. respfig1:** Measurement of the lag in the rise to ½ max between CHMPB and VPS4A in HEK293T cells.

Acidification of the cytosol accelerates scission. My understanding is that the inner surface of the plasma membrane carries a generalized negative charge, so I am not sure how important the PIP_2_ components are in affecting scission. PIP_2_ acts as an important bridge binding the Matrix region of Gag to the inner plasma membrane but this occurs with all the Gag polyproteins from the first stages of assembly even before membrane bending and is not specific to those interactions involved at the budding neck. It would be of interest to see whether formation of vesicles triggered by ESCRT, such as in MVB formation, is similarly affected by pH changes to define how specific this is (or isn't) for virion budding and whether PIP_2_ is of specific importance.

Thank you for the emphasizing the presence of PIP_2_ throughout the Gag bud, and pointing out that PIP_2_ might not be a critical lipid for the scission. Based on the pHluorin fluorescence change we observed when switching between 0% and 10% CO_2_ we believe the cytosolic pH is modulated between ~7.5 and ~6.5. Phosphatidic acid (PA), phosphatidylethanolamine (PE), phosphatidylcholine (PC), phosphatidylglycerol (PG), and phosphatidylserine (PS) all have pK_a_ values that are away from this range. The two phosphates on the inositol of PIP_2_ have a pK_a_ of 6.5 and 7.7, which are in the range of sensitivity of our measurements and therefore we highlighted the possibility of this lipid being sensitive to charge change in the pH range for our experiments. However, we acknowledge that this is just one possible component that is sensitive to charge modulation in this pH range, and we have modified the discussion to more clearly state this possibility.

MVB formation would be an excellent system to also study membrane scission and although outside the scope of this work, we hope our research will provide inspiration for a similar study. In particular, due to differences in lipid composition in MVBs, it would be interesting to determine how scission in MVBs are sensitive to pH.

The demonstration of repeated recruitment and loss of VPS4A and CHMP4B is interesting and novel. It begs the question as to what is the critical trigger for scission as it would appear that although they are important for the process they themselves are necessary but not sufficient. A significant caveat is that the experiments as far as I can tell are performed with Gag alone and there is evidence that the Gag/Pol polyprotein may influence budding significantly with a suggestion that cargo size affects ESCRT function (Bendjennat et al., 2016). Similarly, there is no genomic RNA and this may affect virion assembly. Either of these may be involved in effecting the scission pathway and might be part of the definitive trigger, which would render repeated VPS4A/CHMP4B recruitment unnecessary.

We have previously observed the repeated recruitment and loss of VPS4A and ESCRT components (Jouvenet et al., 2011). We had struggled with the meaning of this until this current work. The results now demonstrate that the durations of ESCRT recruitment and loss are fairly stereotypic in duration. At the end of each round, there is either scission, or not. If there is scission, there are no further rounds of ESCRT recruitment. If there is not scission, then there is another round. We think that the ESCRTs are pulling the neck just an extra bit closer to facilitate scission, but then have to move out of the way. This is repeated until scission occurs. This suggests that it may be possible for scission to occur even in the absence of ESCRTs, albeit at a reduced rate. Thus, we don’t believe there is a definitive trigger for ESCRT-III recruitment. Instead, we hypothesize that the presence of the late domains helps to initiate recruitment of ESCRT-IIIs in a probabilistic fashion and may also be governed by the current neck structure. This is supported by Bendjennat and Saffarian (2016) which showed that the absence of the late domain (deletion of PTAP) delayed particle release by 10 hours but did not prevent viral particle release. Thus, without the late domains, scission events might still be observed, though would be rarer. We have updated the text to include information that scission may occur in the absence of late domain. However, when we previously monitored for scission in the presence of a late domain mutant (Jouvenet et al., 2008), we did not see scission during the 30-60 minute imaging periods we used.

We have previously examined if this assembly is affected by whether Gag or Gag and Gag-Pol was recruited, and at the level of the single virion, the assembly kinetics were the same for Gag alone and for the provirus Gag/Gag-Pol (Jouvenet et al., 2008). We also could not distinguish any differences in assembly rate in the presence of the HIV-1 genome (Jouvenet et al., 2009). The referee points out that in ensemble measurements larger cargo sizes appear to slow down rate of viral particle release (Bendjennat and Saffarian, 2016). We think that this may be an indirect effect seen at the macroscopic level not seen at the level of single virions. Larger cargos take longer to synthesize and slow the overall rate of synthesis in the cell. We have previously demonstrated that rate of viral assembly is very sensitive to concentration of Gag. Gag-Pol takes longer to synthesize than just Gag and thus, it would take longer to get to a critical concentration to initiate assembly and longer to assemble.

I note that nowhere is the role of ESCRT-II mentioned. There is now clear biochemical and EM evidence that it is involved in HIV budding (Meng et al., 2015).

Just as viral particle release rate may depend on presence of ESCRT-I/TSG101 and particle size (Bendjennat and Saffarian, 2016), the presence of ESCRT-II, which works in concert with ESCRT-I, is also expected to affect the rate of viral particle release (Meng et al., 2015) which we have now added to the manuscript.